# Tripartite interactions between filamentous Pf4 bacteriophage, *Pseudomonas aeruginosa*, and bacterivorous nematodes

**Caleb M. Schwartzkopf**[1], **Autumn J. Robinson**[1], **Mary Ellenbecker**[1], **Dominick R. Faith**[1], **Amelia K. Schmidt**[1], **Diane M. Brooks**[1], **Lincoln Lewerke**[2], **Ekaterina Voronina**[1], **Ajai A. Dandekar**[2,3], **Patrick R. Secor**[1] *

1 Division of Biological Sciences, University of Montana, Missoula, Montana, United States of America,
2 Department of Microbiology, University of Washington, Seattle, Washington, United States of America,
3 Department of Medicine, University of Washington, Seattle, Washington, United States of America

* Patrick.secor@mso.umt.edu

**Data Availability Statement:** All data are included in the manuscript and supplemental information.

## Abstract

The opportunistic pathogen *Pseudomonas aeruginosa* PAO1 is infected by the filamentous bacteriophage Pf4. Pf4 virions promote biofilm formation, protect bacteria from antibiotics, and modulate animal immune responses in ways that promote infection. Furthermore, strains cured of their Pf4 infection (ΔPf4) are less virulent in animal models of infection. Consistently, we find that strain ΔPf4 is less virulent in a *Caenorhabditis elegans* nematode infection model. However, our data indicate that PQS quorum sensing is activated and production of the pigment pyocyanin, a potent virulence factor, is enhanced in strain ΔPf4. The reduced virulence of ΔPf4 despite high levels of pyocyanin production may be explained by our finding that *C. elegans* mutants unable to sense bacterial pigments through the aryl hydrocarbon receptor are more susceptible to ΔPf4 infection compared to wild-type *C. elegans*. Collectively, our data support a model where suppression of quorum-regulated virulence factors by Pf4 allows *P. aeruginosa* to evade detection by innate host immune responses.

## Author summary

*Pseudomonas aeruginosa* is an opportunistic bacterial pathogen that infects wounds, lungs, and medical hardware. *P. aeruginosa* strains are often themselves infected by a filamentous virus (phage) called Pf. At sites of infection, filamentous Pf virions are produced that promote bacterial colonization and virulence. Here, we report that strains of *P. aeruginosa* cured of their Pf infection are less virulent in a *Caenorhabditis elegans* nematode infection model. We also report that PQS quorum sensing and production of the virulence factor pyocyanin are enhanced in *P. aeruginosa* strains cured of their Pf infection. Compared to wild-type *C. elegans*, nematodes unable to detect bacterial pigments via the aryl hydrocarbon receptor AhR were more susceptible to infection by Pf-free *P. aeruginosa* strains that over-produce pyocyanin. Collectively, this study supports a model where Pf

**Funding:** PRS is supported by NIH grants R01AI138981 and P20GM103546. AAD is supported by NIH grant R01GM125714 and EV is supported by grant R01GM109053. The funders had no role in study design, data collection and analysis, decision to publish, or preparation of the manuscript.

**Competing interests:** The authors have declared that no competing interests exist.

phage suppress *P. aeruginosa* PQS quorum sensing and reduce pyocyanin production, allowing *P. aeruginosa* to evade AhR-mediated immune responses in *C. elegans*.

## Introduction

Filamentous bacteriophages (phages) of the Inoviridae family infect diverse bacterial hosts [1,2]. In contrast to other phage families, Inoviruses can establish chronic infections where filamentous virions are produced without killing the bacterial host [3–5], which may allow a more symbiotic relationship between filamentous phages and the bacterial host to evolve. Indeed, filamentous phages are often associated with enhanced virulence potential in pathogenic bacteria. For example, the filamentous phage CTXϕ encodes the cholera toxin genes that convert non-pathogenic *Vibrio cholerae* into toxigenic strains [6], the MDAϕ Inovirus that infects *Neisseria gonorrhoeae* acts as a colonization factor and enhances bacterial adhesion to host tissues [7], and the filamentous phage ϕRSS1 increases extracellular polysaccharide production and invasive twitching motility in the plant pathogen *Ralstonia solanacearum* [8].

The filamentous phage Pf4 that infects *Pseudomonas aeruginosa* strain PAO1 enhances bacterial virulence in murine lung [9] and wound [10] infection models. Oxidative stress induces the Pf4 prophage [11] and filamentous virions are produced at high titers, up to $10^{11}$ virions per mL [12,13]. Pf4 virions serve as structural components of biofilm matrices that protect bacteria from antibiotics and desiccation [9,14,15]. Pf4 virions also engage immune receptors on macrophages to decrease phagocytic uptake [10,16] and inhibit CXCL1 signaling in keratinocytes, which interferes with wound re-epithelialization [17]. These observations outline the diverse ways that Pf4 virions promote the initiation and maintenance of *P. aeruginosa* infections. However, how Pf4 phages modulate bacterial virulence behaviors is poorly understood.

*P. aeruginosa* regulates the production of a variety of secreted virulence factors using a cell-to-cell communication system called quorum sensing (QS). As bacterial populations grow, concentrations of QS signaling molecules called autoinducers increase as a function of population density [18]. When autoinducer concentrations become sufficiently high, they bind to and activate their cognate receptors, allowing bacterial populations to coordinate gene expression [18,19]. *P. aeruginosa* PAO1 has three QS systems, Las, Rhl, and PQS. Las and Rhl QS systems recognize acyl-homoserine lactone signals while the PQS system recognizes quinolone signals.

In this study, we demonstrate that deleting the Pf4 prophage from *P. aeruginosa* PAO1 (ΔPf4) activates PQS quorum sensing and increases production of the pigment pyocyanin, a potent virulence factor. However, like observations in vertebrate infection models [9,10], the virulence potential of ΔPf4 is reduced compared to PAO1 in a *Caenorhabditis elegans* nematode infection model. We resolve this apparent controversy and report that *C. elegans* strains lacking the ability to sense bacterial pigments through the aryl hydrocarbon receptor (AhR) are more susceptible to ΔPf4 infection compared to wild-type *C. elegans* capable of detecting bacterial pigments. Collectively, our data support a model where Pf4 suppresses the production of quorum-regulated pigments, allowing *P. aeruginosa* to evade detection by host immune responses.

## Results

### Pf4 protect *P. aeruginosa* from *Caenorhabditis elegans* predation

Prior work demonstrates that Pf4 enhances *P. aeruginosa* PAO1 virulence potential in mouse models of infection by modulating innate immune responses [9,10,16]. Because central

components of animal innate immune systems are conserved, we hypothesized that Pf4 would affect *P. aeruginosa* virulence in other animals such as bacterivorous nematodes. To test this hypothesis, we used *Caenorhabditis elegans* nematodes in a slow-killing *P. aeruginosa* infection model were nematodes are maintained on minimal NNGM agar with a bacterial food source for several days [20].

We first confirmed that PAO1 and ΔPf4 grew equally well on NNGM agar without *C. elegans* (**Fig 1A**) by homogenizing and resuspending three-day-old bacterial lawns in saline and measuring colony forming units (CFUs) by drop-plate. Resuspended cells were then pelleted by centrifugation and Pf4 virions in supernatants were measured by plaque assay. In the absence of *C. elegans*, neither PAO1 nor ΔPf4 produced any detectable Pf4 virions (**Fig 1B**).

Subsequently, we tested the effect of *C. elegans* grazing on PAO1 and ΔPf4. Young adult N2 *C. elegans* were plated onto 24-hour old bacterial lawns and incubated for an additional 48 hours. In the presence of *C. elegans*, PAO1 CFUs were comparable to PAO1 CFUs recovered from lawns grown without *C. elegans* at approximately $10^{10}$ CFUs/mL (**Fig 1C**, black bar, compare to 1A). CFUs recovered from ΔPf4 lawns exposed to *C. elegans* were ~100-fold lower than ΔPf4 lawns grown without *C. elegans* (**Fig 1C**), indicating that Pf4 protects *P. aeruginosa* from *C. elegans* predation.

We did not detect Pf4 virions in ΔPf4 lawns exposed to *C. elegans* (**Fig 1D**), but we did recover ~1 x $10^6$ Pf4 plaque forming units (PFUs) from PAO1 lawns exposed to *C. elegans* (**Fig 1D**, black bar). These results indicate that *C. elegans* predation induces Pf4 virion replication.

When filamentous Pf4 virions accumulate in the environment, they enhance *P. aeruginosa* adhesion to mucus and promote biofilm formation [14,16]. Because *P. aeruginosa* colonization of the *C. elegans* digestive track is a primary cause of death in the slow killing model [20], we hypothesized that Pf4 virions may accumulate in the *C. elegans* digestive track. To test this hypothesis, we topically applied $1 \times 10^9$ fluorescently labeled Pf4 virions to bacterial lawns and imaged *C. elegans* by fluorescence microscopy after 24 hours of grazing. *Escherichia coli* OP50 were used for these experiments to avoid Pf4 replication and any potential bacterial lysis (Pf4 cannot infect *E. coli* hosts). After 24 hours, Pf4 virions accumulated in the upper intestine of *C. elegans* (**Fig 1E**), raising the possibility that Pf4 virions physically block the digestive track, which could increase *C. elegans* killing by *P. aeruginosa*.

When *C. elegans* was challenged with PAO1 in the slow killing model, nematode killing was complete after five days (**Fig 1F** black line) whereas complete *C. elegans* killing took eight days when challenged with ΔPf4 (**Fig 1F** green line), indicating that Pf4 enhances the virulence potential of *P. aeruginosa*, consistent with prior work in mice [9,10,16]. Collectively, these results indicate that *C. elegans* induces Pf4 replication and that Pf4 protects *P. aeruginosa* from *C. elegans* predation.

## PQS quorum sensing is activated and pyocyanin production enhanced in ΔPf4

During routine propagation of *P. aeruginosa*, we noted that production of the green pigment pyocyanin (**Fig 2A**) was significantly (P<0.003) higher in ΔPf4 compared to PAO1 (**Fig 2B and 2C**). Pyocyanin is a redox-active phenazine that shuttles electrons to distal electron acceptors, which enhances ATP production and generates proton-motive force in *P. aeruginosa* cells living in anoxic environments [21,22]. The redox activity of pyocyanin also makes it a potent virulence factor that passively diffuses into phagocytes and kills them by redox cycling with NAD(H) to generate reactive oxygen species that indiscriminately oxidize cellular structures [23].

Expression of many *P. aeruginosa* virulence genes, including the phenazine biosynthesis genes responsible for pyocyanin production, are regulated by quorum sensing [24–31]. We

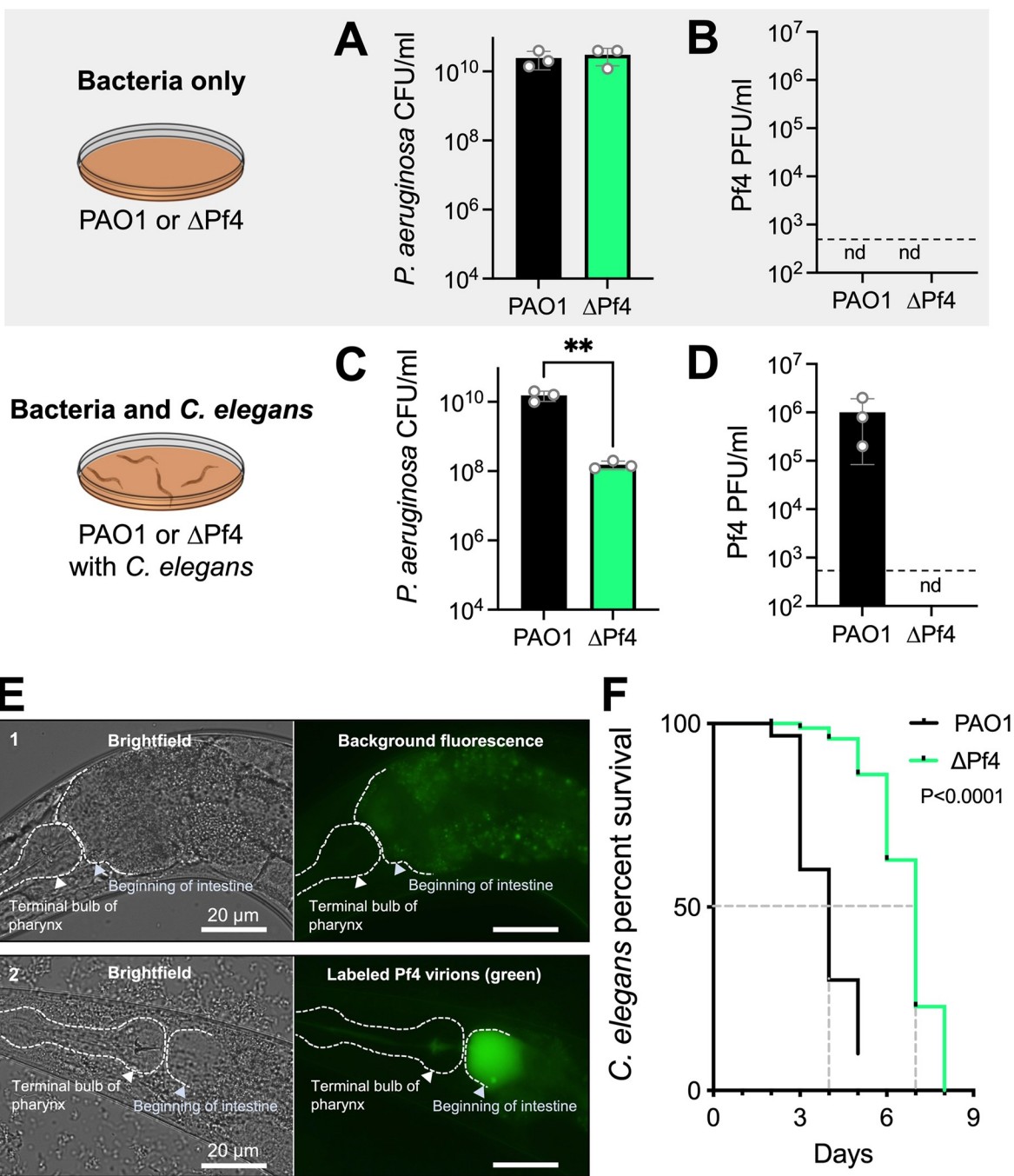

**Fig 1. *C. elegans* predation induces Pf4 replication and enhances *P. aeruginosa* virulence. (A-D)** Bacterial CFUs and Pf4 PFUs were enumerated after three days in the absence (A-B) or presence (C-D) of *C. elegans*. nd, not detected (below detection limit of 333 PFU/mL indicated by dashed line). Results are the mean ±SD of three experiments, **P<0.01, Student's *t*-test. **(E)** Wild-type N2 *C. elegans* were maintained on lawns of 1) *E. coli* OP50 (non-pathogenic nematode food) or 2) OP50 supplemented with 10$^9$ Pf4 virions labeled with Alexa-fluor 488 (green). Representative brightfield and fluorescent images after 24 hours are shown. **(F)** Kaplan-Meier survival curve analysis of *C. elegans* exposed to *P. aeruginosa*. N = 90 worms per condition (three replicate experiments of 30 worms each). The mean survival of *C. elegans* maintained on lawns of PAO1 was four days compared to seven days for nematodes maintained on lawns of ΔPf4 (dashed gray lines). Note that worms that may have escaped the dish rather than died were withdrawn from the study, explaining why the black PAO1 line does not reach zero percent survival.

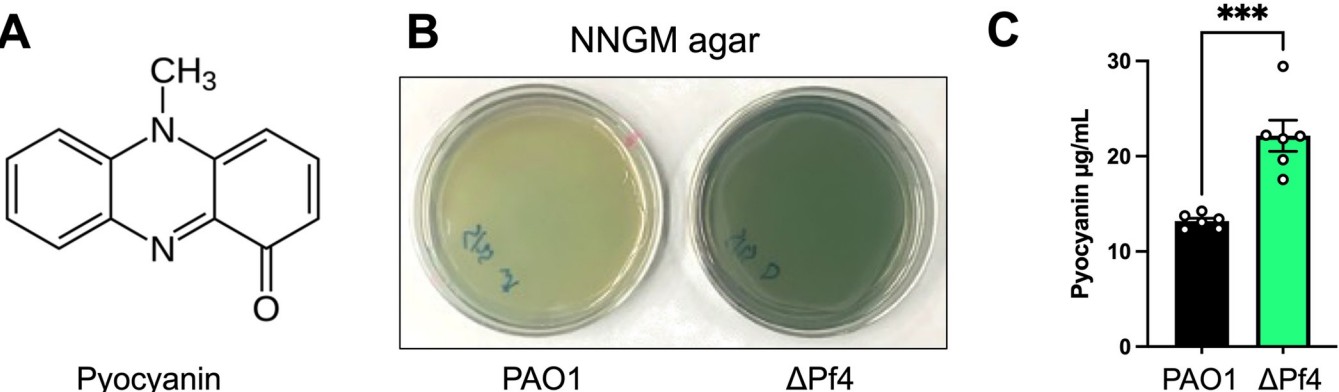

**Fig 2. Pyocyanin production is enhanced in ΔPf4 compared to PAO1. (A)** The structure of pyocyanin, a redox-active green pigment produced by *P. aeruginosa*. **(B)** Representative images of PAO1 and ΔPf4 growing on NNGM agar plates after 24 hours at 37˚C. **(C)** Pyocyanin was chloroform-acid extracted from NNGM agar plates, absorbance measured (520 nm), and values converted to µg/mL. Data are the mean ±SEM of six replicate experiments. ***P<0.003, Student's *t*-test.

used fluorescent transcriptional reporters to measure Las ($P_{rsaL}$::*gfp*), Rhl ($P_{rhlA}$::*gfp*), and PQS ($P_{pqsA}$::*gfp*) quorum sensing [32–34]. In ΔPf4, regulation of Las and Rhl gene targets was not significantly different from PAO1 after 18 hours of growth (**Fig 3A and 3B**). However, PQS activity in ΔPf4 was significantly (P<0.001) higher compared to PAO1 after 18 hours (**Fig 3C**). Fluorescence was not detected in empty vector controls (**Fig 3D**). These results suggest that loss of the Pf4 prophage upregulates PQS quorum sensing, causing pyocyanin to be overproduced.

## Quantitative proteomics analysis of *C. elegans* exposed to PAO1 or ΔPf4

To gain insight into how Pf4 might affect *C. elegans* responses to *P. aeruginosa*, we performed mass spectrometry-based quantitative proteomics on *C. elegans*. To avoid progeny contamination, we used the *rrf-3(−); fem-1(−)* genetic background that is sterile at temperatures above 25˚C [35]. Like wild-type N2 nematodes, PAO1 killed the *rrf-3(−); fem-1(−)* strain significantly (P<0.001) faster than ΔPf4 in the slow killing model (S1 **Fig**). Nematodes were maintained for two days on lawns of PAO1 or ΔPf4. This timepoint was selected because most *C. elegans* were

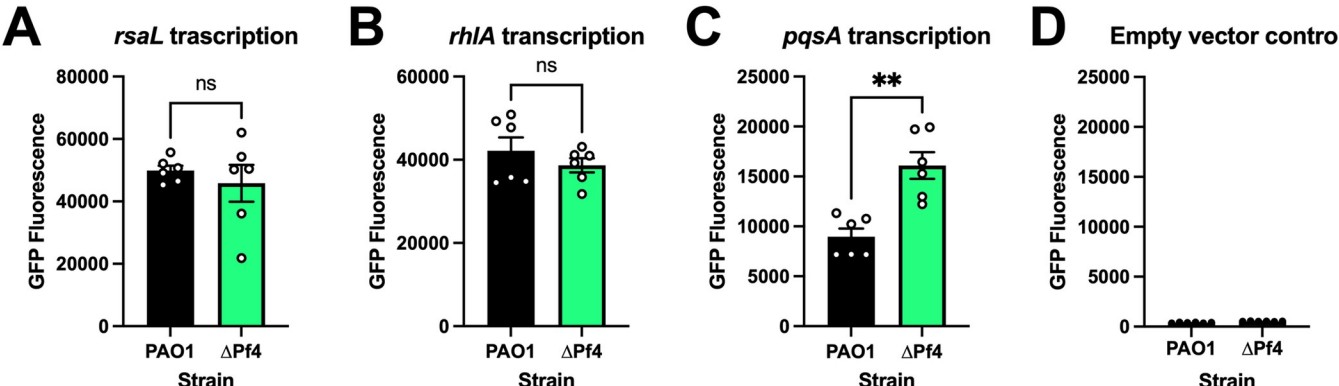

**Fig 3. PQS quorum sensing is upregulated in *P. aeruginosa* ΔPf4.** GFP fluorescence from the transcriptional reporters **(A)** $P_{rsaL1}$-*gfp*, **(B)** $P_{rhlA}$-*gfp*, **(C)** $P_{pqsA}$-*gfp* and **(D)** $P_{empty}$-*gfp* was measured in PAO1 (black) or ΔPf4 (green) at 18 hours in cultures growing in lysogeny broth. For each measurement, GFP fluorescence was corrected for bacterial growth (OD$_{600}$). Data are the mean ±SEM of six replicates. **P<0.001, Student's *t*-test.

still alive in both groups (**Figs 1F and** S1). Whole nematodes were collected (~320 per replicate, N = 4), washed, and proteins purified. Proteins were digested with trypsin and tandem mass tags were used to uniquely label peptides from each biological replicate, allowing all samples to be pooled, fractionated, and analyzed by mass spectrometry in a single run. This approach allows direct and quantitative comparisons between groups.

We identified 410 proteins that were significantly (P<0.05) up or down regulated at least 1.5-fold (log$_2$ fold change ≥0.58) in *C. elegans* exposed to ΔPf4 compared to PAO1 (**Fig 4A and S1 Table**). Enrichment analysis revealed proteins associated with mitochondrial respiration and electron transport were significantly (FDR<0.002) enriched in upregulated proteins (**Fig 4B**). As pyocyanin is a redox-active virulence factor known to interfere with mitochondrial respiration [36,37], these results suggest that respiration is perturbed in *C. elegans* grazing on ΔPf4 lawns that over-produce pyocyanin.

We also noted that proteins associated with muscle cell differentiation and organization were enriched in *C. elegans* challenged with ΔPf4 (**Fig 4C**), which could be related to a decline in motility observed in *C. elegans* as they begin to succumb to *P. aeruginosa* infection [20].

In *C. elegans* exposed to ΔPf4, proteins associated with the extracellular matrix (e.g., collagen) were also significantly enriched (**Fig 4A,** dark blue symbols and **4D**). The tough extracellular cuticle of *C. elegans* is composed predominantly of cross-linked collagen [38]. Because

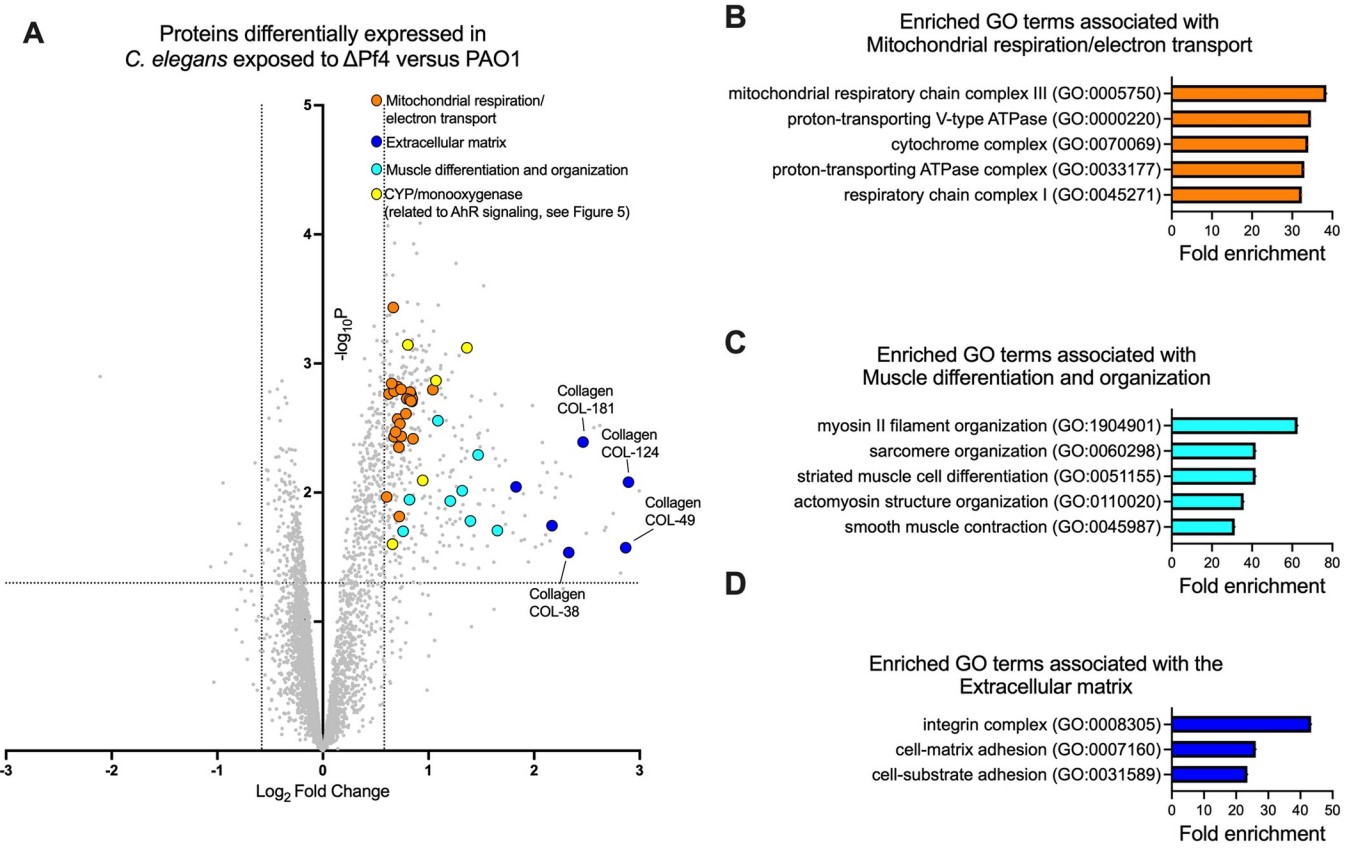

**Fig 4. Pf4 modulates expression of *C. elegans* proteins associated with respiration, the extracellular matrix, and motility.** (A) Volcano plot showing differentially expressed proteins in *C. elegans* maintained on lawns of ΔPf4 compared to *C. elegans* maintained on lawns of PAO1 for three days. The dashed lines indicate proteins with expression levels greater than ±1.5-fold and a false discovery rate (FDR) <0.05. Results are representative of quadruplicate experiments. **(B-D)** Enrichment analysis of significant upregulated proteins shown in (A). Fold enrichment of observed proteins associated with specific Gene Ontology (GO) terms each had an FDR of <0.002.

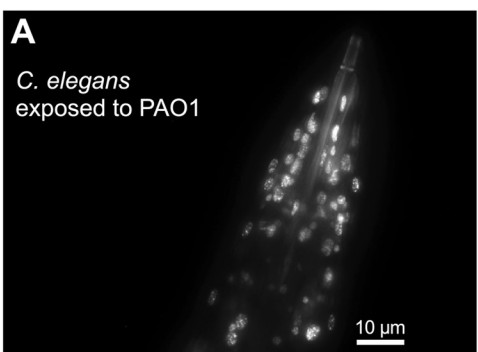
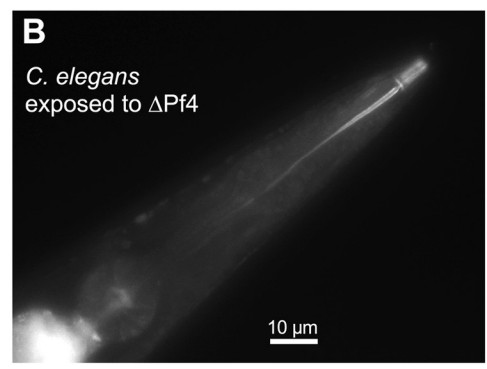
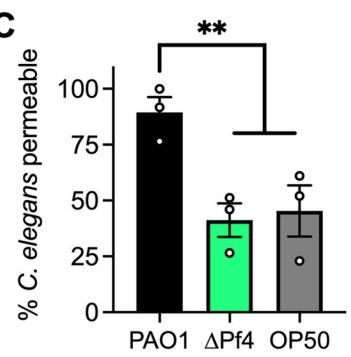

**Fig 5. PAO1 compromises *C. elegans* cuticle integrity compared to ΔPf4.** Synchronized young adult N2 worms were collected from lawns of PAO1, ΔPf4, or *E. coli* OP50 after 48 hours and stained with the nucleic acid stain Hoechst. Cuticle permeability was assessed by visualization of stained nuclei in live nematodes exposed to **(A)** PAO1 or **(B)** ΔPf4. Representative images are shown. **(C)** The percent *C. elegans* with stained nuclei were scored as permeable and plotted. \*\*P<0.01, Student's *t*-test. N = 3 replicates of 25–50 animals per replicate, 92–137 total worms per group.

PAO1 kills *C. elegans* faster than ΔPf4 (**Fig 1F**), lower collagen abundance in PAO1-exposed *C. elegans* may be an indication of compromised cuticle integrity. To test this, we assessed cuticle integrity in synchronized young adult worms collected from lawns of PAO1 or ΔPf4 after two days and stained with 10 μg/mL Hoechst. Nematodes where stained nuclei were observed were scored as permeable and cuticle integrity compromised (**Fig 5A**) whereas worms without stained nuclei were scored as non-permeable with an intact cuticle (**Fig 5B**). We find that *C. elegans* cuticle permeability is significantly (P<0.01) higher in *C. elegans* exposed to PAO1 compared to *C. elegans* exposed to ΔPf4 (**Fig 5C**). These results correlate with the lower relative collagen protein abundance observed in *C. elegans* exposed to PAO1 compared to ΔPf4 (**Fig 4A**) and are consistent with a loss of cuticle integrity and higher morbidity of *C. elegans* exposed *P. aeruginosa* lysogenized by filamentous Pf4 phage.

### *C. elegans* aryl hydrocarbon receptor signaling regulates antibacterial defense

Compared to PAO1, ΔPf4 produces more of the virulence factor pyocyanin (and likely other quorum-regulated virulence factors). However, ΔPf4 is less virulent in mouse lung [9], wound [10], and *C. elegans* infection models (**Fig 1F**). How is it that the ΔPf4 strain that produces more virulence factor is less virulent in animal models of infection?

Prior work demonstrates that vertebrate immune systems can sense *P. aeruginosa* aromatic pigments such as pyocyanin via the aryl hydrocarbon receptor (AhR) pathway [39,40]. AhR is a highly conserved eukaryotic transcription factor that binds a variety of aromatic hydrocarbons and regulates metabolic processes that degrade xenobiotics and coordinate immune responses [39,40]. In vertebrates, AhR's ability to detect pyocyanin and other bacterial pigments provides the host a way to monitor bacterial burden and mount appropriate immune countermeasures [40,41].

Furthermore, AhR regulates the expression of numerous cytochrome P450 (CYP) enzymes in both vertebrates [42] and in *C. elegans* [43] that participate in xenobiotic degradation. In our proteomics dataset, we identified five CYP proteins (CYP-29a2, CYP-25a2, CYP-14a5, CYP-37a1, and CYP-35b1) that were significantly upregulated in *C. elegans* exposed to ΔPf4 (**Fig 4A**, yellow symbols).

Based on these observations, we hypothesized that AhR signaling would increase *C. elegans* fitness against the pyocyanin over-producing ΔPf4 strain. To test this, we challenged wild-type N2 *C. elegans* or an AhR-null mutant (*ahr-1(ia3)*) with PAO1 or ΔPf4 in the slow killing

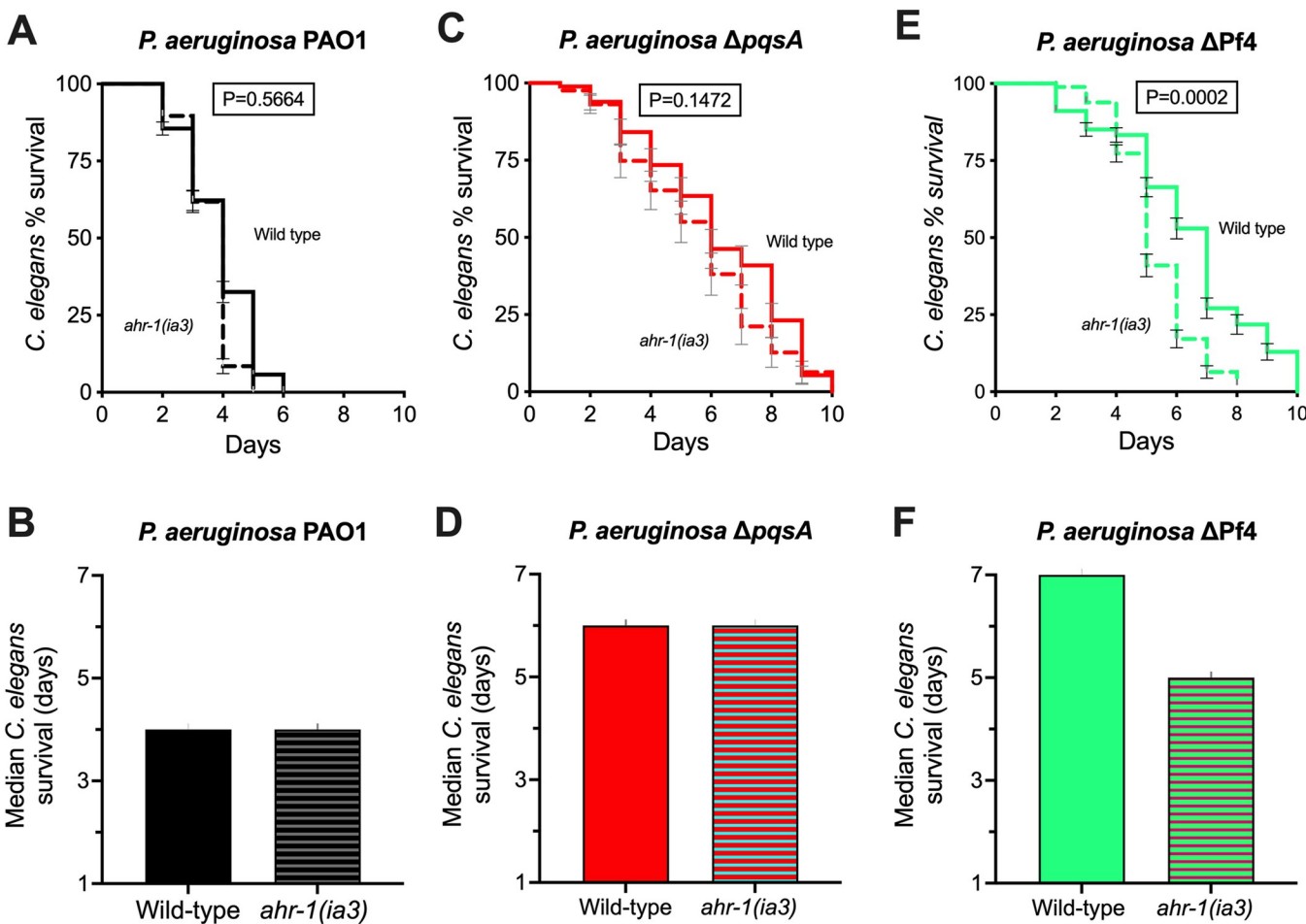

**Fig 6. Inactivation of AhR signaling in *C. elegans* enhances ΔPf4 virulence. (A, C, and E)** Kaplan-Meier survival curve analysis (Log-rank) of wild-type N2 or isogenic *ahr-1(ia3) C. elegans* maintained on lawns of *P. aeruginosa* PAO1, Δ*pqsA*, or ΔPf4 for the indicated times. N = 3 groups of 90 animals per condition (270 animals total per condition). Error bars represent standard error of the mean. P-values of pairwise log-rank survival curve analyses are shown. **(B, D, and F)** The median survival of *C. elegans* in days was plotted for each group.

model. We also included *P. aeruginosa* Δ*pqsA*, a strain where PQS signaling is disabled and pyocyanin production abolished [44]. Δ*pqsA* is far less virulent against *C. elegans* compared to wild-type *P. aeruginosa* (**Fig 6A–6D**, compare black to red), consistent with prior work [45]. Disabling AhR signaling in *C. elegans* does not significantly affect nematode survival when challenged with Δ*pqsA* (**Fig 6C and 6D**). This contrasts with the ΔPf4 mutant where disabling AhR signaling significantly (P = 0.0002) increases ΔPf4 virulence compared to wild-type nematodes (**Fig 6E and 6F**).

These results indicate that even though pigment production is impaired in Δ*pqsA*, additional virulence determinants are inactivated in the Δ*pqsA* mutant. The results also raise the possibility that the Pf4 prophage is targeted in its inhibition of PQS or other pathways that regulate pigment biosynthesis whose products may be sensed by AhR.

## Discussion

Here, we characterize tripartite interactions between filamentous phage, pathogenic bacteria, and bacterivorous nematodes. Our work supports a model where Pf4 phage suppress *P.*

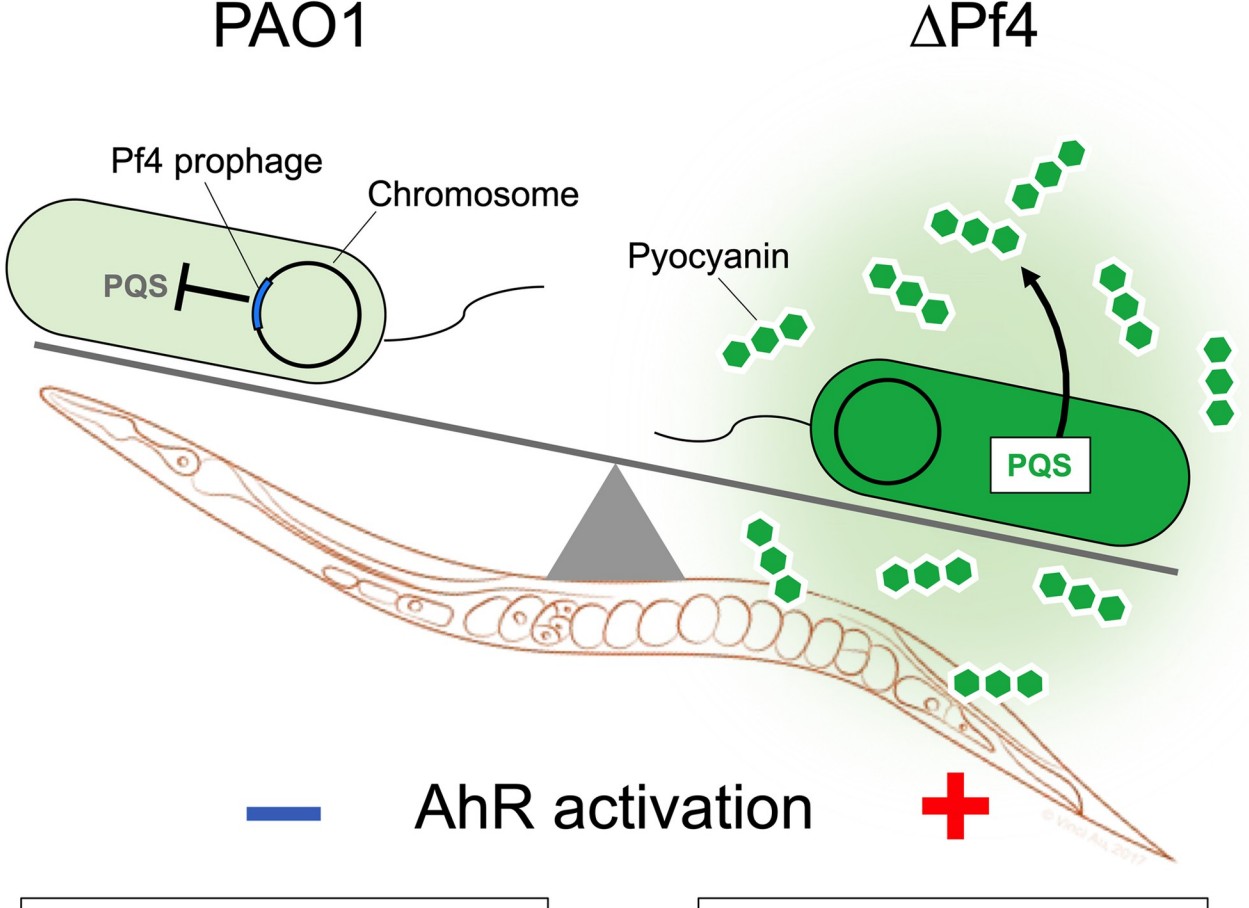

**Fig 7. Proposed model.** Pf4 suppresses the production of quorum-regulated pigments by *P. aeruginosa* allowing bacteria to evade AhR-mediated immune responses in *C. elegans*.

*aeruginosa* PQS quorum sensing and reduce pyocyanin production, allowing *P. aeruginosa* to evade detection by AhR (**Fig 7**).

Many phages modulate bacterial quorum sensing systems [46,47]. Examples in *P. aeruginosa* include phage DMS3, which encodes a quorum-sensing anti-activator protein called Aqs1 that binds to and inhibits LasR [48]. Another *P. aeruginosa* phage called LUZ19 encodes Qst, a protein that binds to and inhibits the PqsD protein in the PQS signaling pathway [49]. In both cases, it is thought that inhibition of *P. aeruginosa* quorum sensing makes the bacterial host more susceptible to phage infection.

Our finding that PQS signaling is upregulated when the Pf4 prophage is deleted suggests that Pf4 encodes proteins that inhibit PQS signaling. The Pf4 prophage encodes a 5' retron element [50] and a 3' toxin-antitoxin pair [51] and these elements may be acting upon host quorum sensing systems. Another possible mechanism involves genes in the Pf core genome as there are still several with unknown function (e.g., *PA0717-PA0720*).

In the absence of *C. elegans*, PAO1 produces significantly less pyocyanin compared to ΔPf4 and infectious Pf4 virions are not simultaneously produced under these conditions. This indicates that the Pf4 prophage can modulate quorum-regulated pigment production during

lysogeny when infectious Pf4 virions are not produced. When *C. elegans* are present, however, Pf4 replication is induced and Pf4 virions appear to accumulate in the *C. elegans* intestine. Pf4 virions are known to promote *P. aeruginosa* biofilm formation and colonization of mucosal surfaces [14,16,52]. It is possible that Pf4 virions may contribute to *P. aeruginosa* colonization of the *C. elegans* intestine, which is a primary cause of *C. elegans* death in the slow killing model [20].

Our study had some limitations. For example, we only measured pyocyanin production by *P. aeruginosa*. Although pyocyanin is often used as an indicator of *P. aeruginosa* virulence potential [53,54], there are many other factors that contribute to *P. aeruginosa* virulence, such as hydrogen cyanide [54]. We also only used well-defined laboratory strains of *P. aeruginosa* and *C. elegans*. While our study suggests that Pf phages may be broad modulators of bacterial virulence, to accurately predict how different *P. aeruginosa* strains (e.g., clinical vs. environmental) might be affected by Pf, future work is required to characterize the effects various Pf strains have on QS systems in different *P. aeruginosa* hosts. One indication that Pf phages may behave differently in various bacterial hosts are variances in QS hierarchies in different *P. aeruginosa* isolates [34]. As quorum sensing can be rewired (e.g., Las dominant verses Rhl dominant hierarchies, [32,55]), it would not be surprising that Pf phage modulate different behaviors in different *P. aeruginosa* hosts.

Our results support a role for AhR signaling in modulating *C. elegans* sensitivity to *P. aeruginosa* infection. Studies in vertebrates reveal that AhR serves as a pattern recognition receptor that senses aromatic bacterial pigments like pyocyanin to initiate appropriate immune responses [39,40]. However, AhR recognizes a diverse array of ligands and modulation of inflammatory responses by AhR is context specific. For example, exposure of airway epithelial cells to combustion products induces pro-inflammatory AhR-dependent responses [56] while activation of AhR by tryptophan metabolites derived from commensal bacteria in the gut is associated with anti-inflammatory responses and maintenance of intestinal barrier integrity [57]. Our proteomics dataset and survival assays suggest that cuticle integrity is compromised in *C. elegans* exposed to PAO1 compared to ΔPf4. An interesting research direction would be to link activation of AhR signaling by bacterial pigments to enhanced cuticle integrity as a potential defense mechanism in nematodes.

In addition to AhR, *C. elegans* has other mechanisms to detect bacterial pigments. In environments illuminated with white light, *C. elegans* can discriminate the distinctive blue-green color of pyocyanin to avoid *P. aeruginosa* [58]. Our studies were performed predominantly in dark environments; future investigations on how Pf4 may affect *C. elegans* spectral sensing of pathogenic bacteria would be interesting. The existence of multiple bacterial pigment detection mechanisms in *C. elegans* highlights the importance of bacterial pigment detection in nematode survival.

Overall, our study provides evidence that Pf4 phage enhance bacterial fitness against *C. elegans* predation. Prior work demonstrates that Pf4 phage also enhance bacterial fitness against phagocytes by inhibiting phagocytic uptake [10,16]. In the environment, nematodes and other bacterivores such as amoeba can impose high selective pressures on bacteria [59–61]. The ability of Pf phage to enhance *P. aeruginosa* fitness against environmental bacterivores may help explain why Pf prophages are so widespread amongst diverse *P. aeruginosa* strains [3,62,63]. The ability of Pf phage to enhance bacterial fitness against bacterivores in the environment may also translate to increased virulence potential in vertebrate hosts, including humans.

## Materials and methods

### Strains, plasmids, and growth conditions

Strains, plasmids, and their sources are listed in **Table 1.** Unless otherwise indicated, bacteria were grown in lysogeny broth (LB) at 37˚C with 230 rpm shaking and supplemented with

**Table 1. Bacterial strains, phage, and plasmids used in this study.**

| Strain | Description | Source |
|---|---|---|
| *Escherichia coli* | | |
| DH5α | Cloning strain | New England Biolabs |
| *P. aeruginosa* | | |
| PAO1 | Wild type | [9] |
| PAO1 ΔPf4 | Deletion of the Pf4 prophage from PAO1 | [9] |
| Bacteriophage Strains | | |
| Pf4 | Inovirus | [14] |
| *C. elegans* | | |
| N2 | Wild type | *Caenorhabditis* Genetic Center |
| ZG24 | AhR null mutant *ahr-1(ia3)* | [67] |
| CF512 | Temperature-sensitive sterile background *rrf-3(b26) II; fem-1(hc17) IV* | [35] |
| Plasmids | | |
| CP59 pBBR1-MCS5 *rsaL-gfp* | GFP *lasI* transcriptional reporter | [34] |
| CP57 pBBR1-MCS5 *rhlA-gfp* | GFP *rhlA* transcriptional reporter | [34] |
| CP53 pBBR1-MCS5 *pqsA-gfp* | GFP *pqsA* transcriptional reporter | [33] |
| CP1 pBBR-MCS5- Blank | GFP empty vector control | [34] |

antibiotics (Sigma) where appropriate. Unless otherwise noted, gentamicin was used at the at either 10 or 30 µg ml$^{-1}$.

## Plaque assays

Plaque assays were performed using ΔPf4 as the indicator strain grown on LB plates. Phage in filtered supernatants were serially diluted 10x in PBS and spotted onto lawns of ΔPf4 strain. Plaques were imaged after 18h of growth at 37˚C. PFUs/mL were then calculated.

## Pyocyanin extraction and measurement

Pyocyanin was measured as described elsewhere [64,65]. Briefly, 18-hour cultures were treated by adding chloroform to a total of 50% culture volume. Samples were vortexed vigorously and the different phases separated by centrifuging samples at 6,000xg for 5 minutes. The chloroform layer (dark blue if pyocyanin present) was removed to a fresh tube and 20% the volume of 0.1 N HCl was added and the mixture vortexed vigorously (if pyocyanin is present, the aqueous acid solution turns pink). Once the two layers were separated, the aqueous layer was removed to a fresh tube and absorbance measured at 520 nm. The concentration of pyocyanin in the culture supernatant, expressed as µg/ml, was obtained by multiplying the optical density at 520 nm by 17.072, as described [65].

## Quorum sensing reporters

Competent *P. aeruginosa* PAO1 and ΔPf4 were prepared by washing overnight cultures in 300 mM sucrose followed by transformation by electroporation [66] with the plasmids CP1 Blank-PBBR-MCS5, CP53 PBBR1-MCS5 *pqsA*-gfp, CP57 PBBR1-MCS5 *rhlA*-gfp, CP59 PBBR1-MCS5 *rsaL*-gfp listed in **Table 1**. Transformants were selected by plating on the appropriate antibiotic selection media. The indicated strains were grown in buffered LB containing 50 mM MOPS and 100 µg ml$^{-1}$ gentamicin for 18 hours. Cultures were then sub-cultured 1:100 into fresh LB MOPS buffer and grown to an OD$_{600}$ of 0.3. To measure reporter fluorescence, each strain was added to a 96-well plate containing 200 µL LB MOPS with a final

bacterial density of $OD_{600}$ 0.1 and incubated at 37˚C in a CLARIOstar BMG LABTECH plate-reader. Prior to each measurement, plates were shaken at 230 rpm for a duration of two minutes. A measurement was taken every 15 minutes for both growth ($OD_{600}$) or fluorescence (excitation at 485–15 nm and emission at 535–15 nm).

### *C. elegans* slow killing assay

Synchronized adult N2, *ahr-1(ia3)*, or *rrf-3(-); fem-1(-) C. elegans* were plated on normal nematode growth media (NNGM) plates with 30 nematodes for each indicated lawn of *P. aeruginosa* and incubated at 30˚C. Over the course of the assay, nematodes were passaged onto new plates of 24-hour-old *P. aeruginosa* lawns daily and counted. Nematodes were counted as either alive or dead with missing nematodes being withdrawn from the study. The study was ended when all nematodes were either dead or missing.

### Preparation of fluorescently tagged Pf4 virions

*P. aeruginosa* ΔPf4 was grown in LB broth to an $OD_{600}$ of 0.5 at 37˚C in a shaking incubator (225 rpm). Five μL of a Pf4 stock containing $5 \times 10^9$ PFU/mL were used to infect the culture. After growing overnight (18h) in the 37˚C shaking incubator, bacteria were removed by centrifugation (12,000 xg, 5 minutes, room temperature) and supernatants filtered through a 0.2 μm syringe filter. Pf4 virions were PEG precipitated by adding NaCl to the filtered supernatants to a final concentration of 0.5 M followed by the addition of PEG 8k to a final concentration of 20% w/vol. After incubating at 4˚C for four hours, the supernatants became noticeably turbid. At this time, phage were pelleted by centrifugation (15,000 xg, 15 minutes, 4˚C), the pellet gently washed in PBS, centrifuged again, and the phage pellet resuspended in 1 mL 0.1 M sodium bicarbonate buffer, pH 8.3. Virions were then labeled with 100 μg of Alexa Fluor 488 TFP ester following the manufacturer's instructions (ThermoFisher). Unincorporated dyes were separated from labeled virions using PD-10 gel filtration columns. PBS was used to elute labeled phages from the column. Titers of labeled phages were measured by qPCR using our published protocol [68]. Labeled phages were aliquoted and stored at -20˚C.

### Fluorescent imaging of nematodes

Approximately $10^9$ Alexa Fluor 488-labeled Pf4 virions in 200 μL PBS were added evenly to 24-hour old *E. coli* OP50 lawns growing on NNGM agar. Plates were incubated at 30˚C for 30 minutes and synchronized adult N2 *C. elegans* were plated. Routine analysis of *C. elegans* by fluorescence/light microscopy was performed after 24 hours by transferring nematodes to a 5% agarose pad containing levamisole (250 mM), a nematode paralytic agent that enables imaging. Nematodes were examined and imaged using a Leica DFC300G camera attached to a Leica DM5500B microscope.

### Protein extraction from *C. elegans*

Proteins were extracted from *rrf-3(-); fem-1(-) C. elegans* as described [69]. Briefly, after *P. aeruginosa* exposure for two days, ~320 *C. elegans* were harvested from NMMG plates into 1.5 mL tubes containing 1 mL PBS. Nematodes were gently mixed by hand, pelleted by centrifugation, and resuspended in 1 mL fresh PBS. *C. elegans* were again pelleted and supernatants were discarded, pellets were weighed and frozen at -80˚C until proteins were ready to be harvested. Pellets were suspended in reassembly buffer (RAB, 0.1M MES, 1mM EGTA, 0.1mM EDTA, 0.5mM $MgSO_4$, 0.75M NaCl, 0.2M NaF, pH7.4) containing Pierce Protease Inhibitor (ThermoScientific, A32965). Samples were sonicated on ice for 10 cycles of a 2 second pulse with 10

seconds rest between pulses. After 2 minutes rest, sonication was repeated for a total of 8 cycles of 10 x 2 second pulses. Lysates were centrifuged at 20,000xg for 30 minutes at 4˚C. Supernatants were transferred to fresh tubes and concentrated to approximately 2μg/μL using 10kDa molecular weight cut off spin columns (VivaSpin 500, Sartorius, VS0102). Protein concentration was determined using a Bradford assay. After visualizing protein integrity by SDS-PAGE (S2A **Fig**), 200 μg total protein for each of the four biological replicates for each treatment were sent to the IDeA National Resource for Quantitative Proteomics Center for proteomic analysis.

## Mass spectrometry-based quantitative proteomics

Total protein (200 μg) from each sample was reduced, alkylated, and purified by chloroform/ methanol extraction prior to digestion with sequencing grade modified porcine trypsin (Promega). Tandem mass tag isobaric labeling reagents (Thermo) were used to label tryptic peptides following the manufacturer's instructions. Labeled peptides were combined into one 16-plex TMTpro sample group that was separated into 46 fractions on a Acquity BEH C18 column (100 x 1.0 mm, Waters) using an UltiMate 3000 UHPLC system (Thermo). Peptides were eluted by a 50 min gradient from 99:1 to 60:40 buffer A:B ratio (Buffer A = 0.1% formic acid, 0.5% acetonitrile. Buffer B = 0.1% formic acid, 99.9% acetonitrile). Fractions were consolidated into 18 super-fractions which was further separated by reverse phase XSelect CSH C18 2.5 um resin (Waters) on an in-line 150 x 0.075 mm column. Peptides were eluted using a 75 min gradient from 98:2 to 60:40 buffer A:B ratio. Eluted peptides were ionized by electrospray (2.4 kV) followed by mass spectrometric analysis on an Orbitrap Eclipse Tribrid mass spectrometer (Thermo) using multi-notch MS3 parameters. MS data were acquired using the FTMS analyzer over a range of 375 to 1500 m/z. Up to 10 MS/MS precursors were selected for HCD activation with normalized collision energy of 65 kV, followed by acquisition of MS3 reporter ion data using the FTMS analyzer over a range of 100–500 m/z. Proteins were identified and quantified using MaxQuant (Max Planck Institute) TMT MS3 reporter ion quantification with a parent ion tolerance of 2.5 ppm and a fragment ion tolerance of 0.5 Da.

## Proteomics data analysis

Prior to data analysis, datasets (**S1 Table**) were subjected to and passed quality control procedures. To assess if there are more missing values than expected by random chance in one group compared to another, peptide intensity values were $Log_2$-transformed (S2B **Fig**). Peptide intensities were comparable across all groups. Principal component analysis (PCA) shows that biological replicates cluster within groups (S2C **Fig**). The normalized $Log_2$ cyclic loess MS3 reporter ion intensities for TMT for the reference *P. aeruginosa* PAO1 proteome (UniprotKB: UP000002438) were compared between wild-type *P. aeruginosa* PAO1 and *P. aeruginosa* PAO1 ΔPf4 conditions. Proteins with $\geq$ 1.5-fold change ($\geq$ 0.58 $log_2$FC) and P values < 0.05 were considered significantly differential. Functional classification and Gene Ontology (GO) enrichment analysis were performed using PANTHER classification system (http://www.pantherdb.org/)) [70]. Analysis results were plotted with GraphPad Prism version 9.4.1 (GraphPad Software, San Diego, CA).

## *C. elegans* cuticle permeability assay

Cuticle integrity was assessed by Hoechst 33342 staining of nuclei in whole nematodes, as previously described [71]. Briefly, synchronized young adult N2 worms were collected from lawns of PAO1 or ΔPf4 after two days and stained with 10 μg/mL Hoechst 33342 for 30 minutes at room temperature. Unbound stain was removed by washing nematodes with M9 buffer before

visualization by fluorescence microscopy using a DAPI filter. Fluorescent images were acquired with a Leica DFC300G camera attached to a Leica DM5500B microscope. All nematodes where stained nuclei were observed were scored as permeable and cuticle integrity compromised.

## Statistical analyses

Differences between data sets were evaluated with a Student's *t*-test (unpaired, two-tailed) where appropriate. P values of $< 0.05$ were considered statistically significant. Survival curves were analyzed using the Kaplan–Meier survival analysis tool. Individual nematodes that were not confirmed dead were removed from the study. The Bonferroni correction for multiple comparisons was used when comparing individual survival curves. GraphPad Prism version 9.4.1 (GraphPad Software, San Diego, CA) was used for all analyses.

## Supporting information

**S1 Fig. Survival analysis of sterile *rrf-3(-); fem-1(-) C. elegans* challenged with *P. aeruginosa* PAO1 or ΔPf4.** Kaplan–Meier survival analysis of N = 90 worms per condition (three replicate experiments of 30 worms each) were monitored daily for death. The mean survival of *rrf-3(-); fem-1(-) C. elegans* maintained on lawns of PAO1 was six days compared to nine days for nematodes maintained on lawns of ΔPf4 (dashed gray lines).
(TIFF)

**S2 Fig. Protein input and proteomics data quality check. (A)** *C. elegans* exposed to PAO1 or ΔPf4 show similar total protein profiles. Forty-five μg of total protein extracted from *C. elegans rrf-3(-); fem-1(-)* exposed to either PAO1 or ΔPf4 for 48 hours was loaded onto a 4–15% Tris Glycine SDS gel and stained with Coomassie blue. Lane 1 Precision Plus All Blue Standard (Bio-Rad 1610373), Lanes 2–5 biological replicates of PAO1 exposed *C. elegans*, Lanes 6–9 ΔPf4 exposed *C. elegans*. Note that after sufficient protein was set aside for mass spectrometry analysis, protein for the sample in lane 9 was limiting, so less was loaded (~35 μg/μL). **(B)** $Log_2$ transformed peptide intensity values were comparable in all datasets. **(C)** Principal component analysis (PCA) shows that biological replicates cluster within groups.
(TIFF)

**S1 Table. Comparative proteomics dataset.**
(XLSX)

## Acknowledgments

We thank Dr. Paul Bollyky and Dr. Laura Jennings for valuable discussions and critical reading of the manuscript. We are grateful to the *Caenorhabditis* Genetics Center for providing *C. elegans* strains and to the IDeA National Resource for Quantitative Proteomics Center at the University of Arkansas.

## Author Contributions

**Conceptualization:** Caleb M. Schwartzkopf, Ekaterina Voronina, Patrick R. Secor.

**Data curation:** Caleb M. Schwartzkopf, Mary Ellenbecker, Diane M. Brooks, Lincoln Lewerke.

**Formal analysis:** Caleb M. Schwartzkopf, Autumn J. Robinson, Mary Ellenbecker, Dominick R. Faith, Amelia K. Schmidt, Diane M. Brooks, Lincoln Lewerke, Ekaterina Voronina, Ajai A. Dandekar, Patrick R. Secor.

**Funding acquisition:** Ajai A. Dandekar, Patrick R. Secor.

**Investigation:** Caleb M. Schwartzkopf, Autumn J. Robinson, Mary Ellenbecker, Dominick R. Faith, Amelia K. Schmidt, Diane M. Brooks, Lincoln Lewerke, Ekaterina Voronina, Ajai A. Dandekar, Patrick R. Secor.

**Methodology:** Mary Ellenbecker, Dominick R. Faith, Amelia K. Schmidt, Diane M. Brooks, Lincoln Lewerke, Ekaterina Voronina.

**Project administration:** Ajai A. Dandekar, Patrick R. Secor.

**Resources:** Ajai A. Dandekar.

**Supervision:** Diane M. Brooks, Ekaterina Voronina, Ajai A. Dandekar, Patrick R. Secor.

**Writing – original draft:** Caleb M. Schwartzkopf, Patrick R. Secor.

**Writing – review & editing:** Caleb M. Schwartzkopf, Ajai A. Dandekar, Patrick R. Secor.

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
