## [Decision Letter · Decision Letter 0]

8 Dec 2022

Dear Dr. Secor,

Thank you very much for submitting your manuscript "Tripartite interactions between filamentous Pf4 bacteriophage, Pseudomonas aeruginosa, and bacterivorous nematodes" for consideration at PLOS Pathogens. As with all papers reviewed by the journal, your manuscript was reviewed by members of the editorial board and by several independent reviewers. The reviewers appreciated the attention to an important topic. Based on the reviews, we are likely to accept this manuscript for publication, providing that you modify the manuscript according to the review recommendations.

Please address all major and minor issues raised by the reviewers. Both reviewers expressed concern regarding the significance of the proteomics. From my perspective the proteomics adds value to the study, but a more directed follow up (such as that suggested by reviewer #2) would increase the impact of the manuscript.

Sincerely,

Matthew C Wolfgang

Academic Editor

PLOS Pathogens

David Skurnik

Section Editor

PLOS Pathogens

Kasturi Haldar

Editor-in-Chief

PLOS Pathogens

orcid.org/0000-0001-5065-158X

Michael Malim

Editor-in-Chief

PLOS Pathogens

orcid.org/0000-0002-7699-2064

Please address all major and minor issues raised by the reviewers. Both reviewers expressed concern regarding the significance of the proteomics. From my perspective the proteomics adds value to the study, but a more directed follow up (such as that suggested by reviewer #2) would increase the impact of the manuscript.

Reviewer Comments (if any, and for reference):

Reviewer's Responses to Questions

**Part I - Summary**

Reviewer #1: Review of Tripartite interaction between filamentous Pf4 bacteriophage, Pseudomonas aeruginosa, and bacterivorous nematodes.

Schwartzkopf and colleagues have written a well structed study into the interaction between QS Pf4 and C elegans and PAO1. We are presented the observation that Pf4 cured strains (∆Pf4) are less virulent than wildtype even though they typically produce higher levels of a known virulence factor pyocyanin. This counter intuitive claim is followed by an examination of the model (c elegans) immune response to virulence factors. Overall, I think this paper should be published and the findings are of broad interests to several disciplines. I have some minor comments which I think would improve the paper and a couple of minor mistakes which need addressing.

Reviewer #2: In this work, Schwartzkopf et al. demonstrate mechanisms by which bacteriophage Pf4 influences Pseudomonas aeruginosa virulence. Using C. elegans as a model, they show that WT PAO1 is more virulent than strains that have been cured of their Pf4 infection (DPf4), despite DPf4 overproduction of pyocyanin. They demonstrate mutant C. elegans with impaired pigment detection are more susceptible to DPf4, suggesting a mechanism of resistance in WT C. elegans. This work also supports their proposed model of Pf4 expression playing a role in the suppression of PQS quorum sensing and subsequent expression of pyocyanin.

This work was well designed and presented. A major strength of the paper was that it also examined infection from a host perspective, using proteomics and gene ontology to quantify protein expression due to infection and extrapolate possible underlying mechanism. This data is well presented in Figure 4. Their overall proposed model is nicely summarized in Figure 6, which is another major strength of the paper.

**Part II – Major Issues: Key Experiments Required for Acceptance**

Reviewer #1: Comments

Could the authors defend why they used rsaL, rhlA and pqsA? Why not lasR or rhlR for example? I can think of several reasons but it would be good to have the authors state theirs.

I am unsure of the value of the proteomic section. I think the results are well presented and clearly there is a difference between the conditions, I am not sure what value it adds to the publication. It does not spur the work with AhR as far as I can tell. This section could be removed without hurting the conclusions. If I have missed a key connection, please make it more obvious.

The Pf4 mechanism for interaction with PQS on page 12 could do with some citations as to why the authors think these factors might interact.

I felt that the mention of medical implants and Cystic fibrosis where a little tacked on. Several of the authors have done fantastic work on QS in CF and Pf4 in CF, I would love to hear their expanded thoughts of how this work might be digested by the fields.

Finally, I would really like to see the results in figure 5 reproduced with a pyocyanin knockout as well as bacteria load calculations. It would be interesting to know if the bacterial load is constant and the immune response is responding differently. However I understanding asking for additional experimentation can be highly difficult so I am open to arguments.

Reviewer #2: Although the proteomics data is very informative, it would be strengthened further by additional assessment of the cuticle of the C.elegans model. For example, if there is a way to correlate protein abundance (lowered in higher morbidity worms) with actual integrity of the protein. Visual evidence of the breakdown of the cuticle with this prediction would be very useful.

**Part III – Minor Issues: Editorial and Data Presentation Modifications**

Reviewer #1: Minor mistakes:

Line 117 the words feed on are missing in brackets.

Line 188 Did the authors mean to just write “Methods”?

(very minor) In figures 1-3 the authors use *,**,*** denotations for significance. I’m of the opinion that significance is binary. Being more significant should not be a consideration.

Reviewer #2: Minor revisions:

-Line 97: missing a delta from “In the absence of C. elegans, neither PAO1 nor DPf4 produced any detectable..”

-Line 117: missing “infect” in Pf4 cannot ____ E. coli hosts

PLOS authors have the option to publish the peer review history of their article (what does this mean?). If published, this will include your full peer review and any attached files.

Reviewer #1: **Yes: **James Gurney

Reviewer #2: No

Figure Files:

Data Requirements:

Reproducibility:

References:

---

## [Editor Report · Decision Letter 1]

8 Feb 2023

Dear Dr. Secor,

We are pleased to inform you that your manuscript 'Tripartite interactions between filamentous Pf4 bacteriophage, Pseudomonas aeruginosa, and bacterivorous nematodes' has been provisionally accepted for publication in PLOS Pathogens.

Best regards,

Matthew C Wolfgang

Academic Editor

PLOS Pathogens

David Skurnik

Section Editor

PLOS Pathogens

Kasturi Haldar

Editor-in-Chief

PLOS Pathogens

orcid.org/0000-0001-5065-158X

Michael Malim

Editor-in-Chief

PLOS Pathogens

orcid.org/0000-0002-7699-2064
---

## [Editor Report · Acceptance letter]

15 Feb 2023

Dear Dr. Secor,

We are delighted to inform you that your manuscript, "Tripartite interactions between filamentous Pf4 bacteriophage, Pseudomonas aeruginosa, and bacterivorous nematodes," has been formally accepted for publication in PLOS Pathogens.

Best regards,

Kasturi Haldar

Editor-in-Chief

PLOS Pathogens

orcid.org/0000-0001-5065-158X

Michael Malim

Editor-in-Chief

PLOS Pathogens

orcid.org/0000-0002-7699-2064